# Effects of Emotional Regulation, Resilience, and Distress Disclosure on Post-Traumatic Growth in Nursing Students

**DOI:** 10.3390/ijerph20042782

**Published:** 2023-02-04

**Authors:** Kyungmi Kim, Jongeun Lee, Jaeyeon Yoon

**Affiliations:** 1Department of Nursing, Gangdong University, Eumseong-gun 27690, Republic of Korea; 2Department of Nursing Science, College of Medicine, Chungbuk National University, Cheongju-si 28644, Republic of Korea

**Keywords:** PTG, emotional regulation, resilience, distress disclosure, nursing students

## Abstract

Nursing students, who need to reflect on self, secure their identity, and be prepared as would-be nurses, can make a good use of post-traumatic growth (PTG) that can function as a catalyst for positive change even amidst this COVID-19 crisis. Emotional regulation strategies in traumatic events are key factors for successful growth, resilience is positively associated with PTG, and distress disclosure is an important factor for stress reduction. In this context, this study is a descriptive research study to identify factors influencing the PTG of nursing students, using emotional regulation, resilience, and distress disclosure as the main variables. Data were collected from 231 junior and senior students of the nursing departments of two universities, and the collected data were analyzed using the t-test, the Mann–Whitney U test, ANOVA, the Scheffé test, Pearson’s correlation coefficients, and stepwise multiple regression in SPSS/WIN 26.0. Analysis of the PTG scores of the nursing students by general characteristics revealed significant differences in PTG according to the transfer status, perceived health status, and levels of satisfaction with major, hybrid-learning class, interpersonal relationship satisfaction, and clinical practice. Factors influencing PTG were identified to be resilience, reappraisal among emotional regulation strategies, satisfaction with clinical practice, and transfer, with the overall explanatory power calculated at 44%. Based on the results of this study, it is necessary to consider resilience and reappraisal, which is a sub-variable of emotional regulation strategies, in order to develop programs designed to promote PTG of nursing students in the future.

## 1. Introduction

COVID-19, the worst global public health crisis ever, with its unprecedented transmission rate and mutation process that defy comparison with previous epidemics, such as SARS or MERS, has a tremendous impact on our daily lives and behavior, causing depression and extreme stress [1]. People who have been exposed directly or indirectly to the COVID-19 pandemic over the past two years may develop posttraumatic stress disorder (PTSD) symptoms [2]. PTSD is a mental illness that appears after experiencing extreme stress that threatens life of body [2]. Such psychosocial effects can appear and persist in complex forms of physical sequelae and psychological problems such as anxiety, depression, and PTSD symptoms, and social problems such as unemployment and economic difficulties, even after the end of the pandemic [3]. In a study analyzing the impact of the lockdown policy due to the COVID-19 pandemic on the mental health of the general public, PTSD symptoms were observed, more markedly in women and younger adults [4]. However, while traumatic events can trigger PTSD symptoms, they can also act as a catalyst for positive change [5].

Post Traumatic Growth (PTG) refers to positive psychological changes perceived by individuals after exposure to traumatic or adverse events by achieving a personal development that goes beyond their usual psychological functions and coping abilities. It means a change in human psychology [6]. Trauma is a wound caused by experiencing a shocking and painful event [7]. Traumatic events include physical and emotional maltreatments and disasters, including war [8], and the South Korean “Framework Act on Disaster and Safety Management” defines infectious diseases as social disaster [9]. Thus, COVID-19-like disaster events can have considerable adverse effects on individuals [10]. However, traumatic experiences do not necessarily develop into maladjustment or psychopathology; rather, they can also function as a driving force to further growth [11].

Over the lifetime, 71.9% of adolescents, college students, and adults experience one or more traumatic events [12], and 26.5% of nursing students were found to have experienced trauma in their daily lives, and 61.2% during clinical practice [13]. Prior to COVID-19, most of the nursing education courses in Korea were face-to-face classes, except for practical training, but after the COVID-19 pandemic, various forms of hybrid learning were introduced. Hybrid learning is a teaching method where teachers instruct in-person and remote students at the same time [14]. Nursing students are required to have the ability to flexibly cope with these types of classes. Nursing students perceive fear of COVID-19 infection and death and can lose confidence and fall into depression due to severance of interpersonal relationships, economic difficulties, fear of practice, and fear of coursework and employment. However, they were also found to begin to reflect on self amid this global crisis, secure their identity, and realize their role as student nurses [15]. Emotional regulation in traumatic events is a key factor for successful growth, resilience is positively associated with life satisfaction and PTG [16], and distress disclosure is an important factor for stress reduction [17].

Emotional regulation is an individual’s ability to react to stress as a coping strategy that regulates a wide range of functions such as emotional, behavioral, cognitive, physiological, and environmental functions [18], and has been attracting increasing interest as an important factor in relation to social adaptation problems [19]. Emotional regulation in nursing students was found to be conducive to stress reduction and help improve their nursing professionalism [20]. Emotional dysregulation was found to be a contributing factor for the exacerbation of PTSD symptoms [21], and emotional regulation is essential in effectively alleviating and preventing PTSD symptoms in college students exposed to trauma [22].

Resilience can function as an important growth factor in individuals because it reflects the ability to return their energies and abilities impaired through exposure to trauma back to the pre-trauma levels and to achieve recovery [23]. Nursing students’ ability to overcome difficult situations and endure adversity is attributable to resilience [24]. Nursing students’ resilience is a predictor of their ability to cope with stress, and those with high resilience can leverage positive emotions for recovery even in the face of stress [25].

Distress disclosure refers to an individual’s tendency to reveal personally distressing information to others [26]. Individuals who can express their feelings well, verbally or non-verbally, can reduce their pain caused by suppressed emotions and improve their insights, which has a desirable effect on interpersonal relationships [27]. Nursing students’ self-disclosure has been reported to be a major factor in reducing clinical practice stress [17]. In a related study, participants who proceeded to deep self-disclosure to a person who listened to and accepted their pain reported on their feeling of relief and lightness of heart [28]. Disclosing one’s painful events and negative emotions to a social supporter promotes PTG through its effect of alleviating negative feelings [29]. In addition, some studies [30,31] demonstrated that distress disclosure was associated with PTG.

People with traumatic experiences try to understand themselves and the world anew by examining the causal relationships and meaning of the problem after the traumatic experience. This challenge, which is a process of emotional regulation of severe stress caused by the traumatic experience, helps trauma survivors better adapt and promotes PTG [32]. There are studies on nursing students’ traumatic experiences and PTG during clinical practice [13] and the experience of PTG experience in nursing students [28]. However, little research has been dedicated to PTG after the national disaster of the COVID-19 pandemic. Therefore, this study was conducted to examine the PTG factors in nursing students who had experienced the trauma of the COVID-19 pandemic with the aim of preparing basic data to help nursing students better adapt. To this end, the following objectives have been formulated:

(1) Examining the differences in PTG according to the general characteristics of nursing students;

(2) Identifying the emotional regulation strategies, resilience, distress disclosure, and PTG level in nursing students;

(3) Calculating the correlations between major variables;

(4) Determining the factors influencing PTG in nursing students.

## 2. Materials and Methods

### 2.1. Study Design and Participants

The present study used a descriptive survey design to identify the relationship between emotional regulation, resilience, distress disclosure and PTG in nursing students and to identify the factors influencing PTG. The sample size was calculated using the G power 3.1 program. Based on a medium effect size of 0.15, a significance level of 0.05, a power of 0.90 and 21 predictors used for multiple linear regression analysis, the minimum number of participants required for this study was 195. The study population consisted of 231 nursing students (115 and 116 juniors, and seniors, respectively) from two nursing colleges in Chungcheong Province, Korea.

### 2.2. Research Instruments

#### 2.2.1. Post Traumatic Growth Inventory-X (PTGI-X)

The level of PTG was measured using the Post Traumatic Growth Inventory-X (PTGI-X) developed by Tedeschi et al. [33] and adapted into Korean and validated by Kim et al. [34]. This questionnaire consisted of a total of 25 items, with each item rated on a 6-point Likert scale (1 = not at all, 6 = to a very great degree). The total score is calculated by adding up the item scores, and a higher score indicates a higher level of positive posttraumatic changes. The internal reliability (Cronbach’s alpha) was 0.97 in the study by Kim et al. [34], and 0.95 in this study.

#### 2.2.2. Emotional Regulation Questionnaire (ERQ)

Emotional regulation strategy was measured using the Emotional Regulation Questionnaire (ERQ) developed by Gross and John [35], adapted by Shon [36], and modified by Cho and Lee [37] into an easy-to-answer 5-point scale. This 10-item psychometric instrument consists of two domains: reappraisal (6 items) and suppression (4 items). Each item is rated on a 5-point Likert scale (1 = not at all, 5 = to a very great degree), with a higher total score indicating a higher degree of using the emotional regulation strategy of cognitive reassessment or emotional suppression. The internal reliability of ERQ was 0.85 for reappraisal and 0.73 for suppression in the study by Shon [36], and 0.87 and 0.77, respectively, in this study.

#### 2.2.3. Resilience

Resilience was measured using a resilience scale developed by Yang et al. [38] for nursing students. Each item of this 24-item instrument is rated on a 5-point Likert scale (1 = not at all, 5 = to a very great degree), with a higher total score indicating a higher level of resilience. The internal reliability of the instrument was 0.84 at the time of development, and 0.90 in this study.

#### 2.2.4. Distress Disclosure Index (DDI)

The level of distress disclosure was measured using the Distress Disclosure Index (DDI) developed by Kahn and Hessling [39] and adapted by Shin and Ahn [40]. DDI measures an individual’s tendency to disclose emotions and thoughts after experiencing a negative event. Each item of this 12-item instrument is rated on a 5-point Likert scale (1 = not at all, 5 = to a very great degree), with a higher total score indicating a higher tendency to disclose uncomfortable information. The internal reliability (Cronbach’s alpha) of the original scale was 0.95, that of the adapted version as reported by Shin and Ahn [40] was 0.92, and 0.92 in this study.

### 2.3. Data Collection and Ethical Consideration

The present study was conducted with an approval from the Institutional Review Board of “C” University. Data were collected between 20 and 26 June 2022, using Google Forms as a non-face-to-face mobile questionnaire survey. To protect personal information, items containing personal identifiable information were not separately collected, and serial numbers were assigned to the questionnaire to ensure confidentiality. The announcement of recruitment of research participants was posted on the social media platform used by all university students, along with a link to the Google Form for the online consent form and information sheet. The questionnaire required approximately 15 min to complete. After completing the survey, we provided mobile drink coupons to those who provided their personal contact information as a token of appreciation for their participation. After the coupon was issued, all personal information was destroyed.

### 2.4. Data Analysis

Data collected in the study were analyzed using the IBM SPSS Statistics 26.0 program (IBM Corp, NY, USA). The general characteristics of the participants were analyzed using frequency/percentage and mean/standard deviation (SD). Analysis of the differences in PTG according to general characteristics was performed using the independent t-test, one-way ANOVA, and the Mann–Whitney test, and the Scheffé test was used for post hoc testing. The correlations between PTG, emotional regulation, resilience, and distress disclosure were calculated using Pearson’s correlation coefficients, and a stepwise regression analysis was performed to determine the factors influencing the participants’ PTG.

## 3. Results

### 3.1. Respondents’ General Characteristics

Respondents’ general characteristics can be summarized as follows: approximately equal number of juniors (49.8%) and seniors (50.2%) participated in the survey; gender distribution was uneven with 15.2% males and 84.8% females; transfer students accounted for 5.6%; the majority of students were living with the family (65.4%), had no religions (68.0%), and rated their parents’ economic status as middle level (53.2%); approximately equal number of students rated their health status as healthy (45.9%) or average (46.3%), was satisfied (46.8%) or neutral (45%) with their major, and was neutral (46.8%) or dissatisfied (41.6%) with their academic achievements; “neutral” was the most frequent response to the question about satisfaction with hybrid-learning class (47.2%), interpersonal relationship (52.8%), and clinical practice (56.3%) (Table 1).

The results of analyzing the differences in PTG according to general characteristics are as follows. PTG showed significant differences depending on the transfer status (t = −2.14, *p* = 0.032), as did the subjective health status, with the group that perceived themselves as “healthy” scoring significantly higher in PTGI-X than the “poor” health group in post hoc testing (F = 3.37, *p* = 0.036). Differences were observed in PTG between different levels of satisfaction with the major, with the “satisfied” ‘and “neutral” groups scoring significantly higher than the “dissatisfied” group in post hoc testing (F = 8.28, *p* < 0.001). Differences were observed also in PTG between different levels of satisfaction with the hybrid-learning class, with the “satisfied” group scoring significantly higher than the “dissatisfied” group in post hoc testing (F = 4.13, *p* = 0.017). Differences were observed also in PTG between different levels of satisfaction with interpersonal relationship, with the “neutral” and “satisfied” groups scoring significantly higher than the “dissatisfied” group in post hoc testing (F = 12.70, *p* < 0.001). Significant differences were observed in PTG between different levels of satisfaction with clinical practice, scoring in the PTGI-X in descending order or satisfied, neutral, and dissatisfied groups in post hoc testing (F = 12.63, *p* < 0.001).

### 3.2. PTG, Emotional Regulation, Resilience, and Distress Disclosure Levels

The participants’ overall mean PTG was 3.90 ± 0.91. The mean scores of the two domains of emotional regulation were 2.61 ± 0.92 and 3.32 ± 0.87 for suppression and reappraisal, respectively. The mean score for resilience was 3.70 ± 0.54, and that of distress disclosure, 3.25 ± 0.81 (Table 2).

### 3.3. Correlations among PTG, Emotional Regulation, Resilience, and Distress Disclosure

Table 3 shows the analysis results of the correlation between PTG, emotional regulation, and distress disclosure of the subjects. The participant’s PTG showed significant positive correlations with the emotional regulation strategy “reappraisal” (r = 0.56, *p* < 0.001), resilience (r = 0.59, *p* < 0.001), and distress disclosure (r = 0.18, *p* = 0.005). The emotional regulation strategy “suppression” showed a significant positive correlation with the emotional regulation strategy “reappraisal” (r = 0.23, *p* < 0.001) and significant negative correlation with distress disclosure (r = −0.62, *p* < 0.001). Resilience showed a significant positive correlation with distress disclosure (r = 0.32, *p* < 0.001).

### 3.4. Determinants of PTG

Factors influencing nursing students’ PTG were analyzed using stepwise regression, with the variables that exhibited significant differences in the participants’ general characteristics, resilience, emotional regulation strategy “reappraisal,” and distress disclosure set as independent variables. Among these variables, categorical variables were entered into the regression model after being converted to dummy variables. Consequently, the regression model was found to be statistically significant (F = 45.24, *p* < 0.001), with its explanatory power calculated at 44%. The variance inflation factor (VIF) ranged from 1.01 to 1.66, with all variables determined to have no multicollinearity problem. Analysis of the residuals confirmed the normality of error terms, homogeneity of variances, and the linearity of the model. In the autocorrelation test, the Durbin-Watson statistic stood at 1.97, i.e., close to 2, indicating the absence of autocorrelation. The most important factor influencing PTG was resilience (β = 0.37, *p* < 0.001), followed by emotional regulation strategy “reappraisal” (β = 0.31, *p* < 0.001), satisfaction with clinical practice (β = 0.14, *p* = 0.005), and transfer status (β = 0.13, *p* = 0.012) (Table 4).

## 4. Discussion

This study examined the correlations among PTG, emotional regulation, resilience, and distress disclosure in nursing students, and determined the factors influencing their PTG. As a result of this study, the factors influencing PTG were resilience and emotional regulation strategy—’reappraisal’, satisfaction with clinical practice, and transfer status.

The participants’ general characteristics that showed differences in PTG were transfer status, subjective health status, and the levels of satisfaction with major, hybrid-learning class, interpersonal relationship, and clinical practice.

Transfer students scored higher in PTG than general admission students. The PTG score of transfer students was higher compared to the results of previous studies on nursing students [13,41]. The Department of Nursing is a department that operates a nursing education certification program and requires additional credits for graduation in addition to the graduation requirements of the university. Therefore, in the case of general transfer, even if transferred to the third year, it is common for more than one year of study time to be added to graduate, and students who want to transfer apply with this burden. Although a direct comparison cannot be made as there is no similar study, it is thought that the high PTG score of transfer students is due to the challenge and effort to transfer to the department of nursing while putting down the existing stable life [42].

Nursing students who perceived their health as good and were satisfied with clinical practice scored higher in PTG compared with those who answered otherwise, which is consistent with the findings of previous research [13]. A positive perception of one’s health can maintain and improve physical and mental health and enhance access to social support [43], which in turn may contribute to enhancing PTG.

In this study, nursing students who were dissatisfied with their major showed lower PTG than those who were not, consistent with the result of a previous study [41]. Nursing students dissatisfied with their major can have confused and negative views of their job, which can negatively affect their adaptation to clinical settings [44]. Low satisfaction with major leads to low self-esteem, which in turn can exert a negative effect on PTG [45].

Students who responded flexibly to the hybrid-learning class scored higher in PTG than those who did not. In the context of the COVID-19 pandemic, students participated in various types of classes in which real-time remote classes and asynchronous video classes were mixed. While a direct comparison cannot be made for lack of previous research on the relationship between satisfaction with hybrid-learning class and PTG, it is worthwhile to note the results of a study on college students attending e-learning classes [46] that the level of emotional regulation varied depending on the level of participation in e-learning activities and that passive learners experienced negative emotions more frequently than active learners. These results support the research finding the active participation in e-learning classes help the learner vent their feelings instead of suppressing them, which can positively affect PTG [47]. Therefore, it is necessary for teachers to use various teaching methods to better motivate learners to actively participate in class.

Nursing students satisfied with their interpersonal relationships scored higher in PTG than their dissatisfied counterparts, which is consistent with the research finding that PTG scores were significantly higher in university students who were satisfied with their interpersonal relationships than who were not [48], and also in line with the research finding that social support is needed to succeed in PTG [49].

Finally, the positive correlation between PTG and satisfaction with clinical satisfaction is also in line with the research finding that nursing students with high satisfaction with clinical practice can actively cope with stress and adapt to crisis situation [50,51].

The PTG score of the nursing students in this study was 78 on a 100-point scale, which is higher than the 61.3 points [41] and 52.6 points [13] of the studies using the original version of PTGI prior to expansion. From the higher score achieved in this study using the PTGI-X, the extended version of PTGI, it can be inferred that the expanded version measures more sensitively than the original version by adding four existential growth items. The resilience score of this study was 3.70 ± 0.54, which was slightly higher than 3.56 ± 0.44 of a previous study on nursing students [52]. The mean score for suppression, one of the two domains of emotional regulation, was 2.61 ± 0.92, which is slightly higher than that of 2.56 ± 4.53 achieved by health teachers [53] and similar to 2.62 ± 2.91 achieved by adolescents [54]. The mean score for reappraisal, the other domain of emotional regulation, was 3.32 ± 0.87, lower than 3.59 ± 9.37 and 3.58 ± 3.86 achieved by health teachers and adolescents [53,54]. This indicates that nursing students are less successful than health teachers and adolescents in changing their thoughts as an emotional regulation strategy. The nursing students’ mean score for distress disclosure was 3.25 ± 0.81, similar to 3.25 ± 8.93 in college students [55] and slightly lower than 3.36 ± 0.48 in firefighters [30].

According to the correlation analysis of this study, nursing students’ PTG was positively correlated with the emotional regulation domain “cognitive reappraisal,” but not with the other domain “emotional suppression,” which is consistent with the research finding that the higher the cognitive reappraisal, the higher the PTG score in college students [56], but inconsistent with the research finding that adults who frequently used emotional suppression strategy did not experience PTG [57]. This may be interpreted as meaning that cognitive reappraisal may have a stronger impact on PTG compared to emotional suppression by significantly reducing discomfort [58,59,60]. Therefore, to help nursing students better succeed in PTG, it is necessary to provide a program with contents that can promote cognitive reappraisal. Nursing students’ resilience was found to be correlated with PTG and determined as a contributing factor for PTG, which is consistent with the reports in previous studies [41,61,62,63]. Since resilience is an intrapersonal factor prior to trauma, confirmed to directly promote PTG, it is necessary to strengthen resilience as a preventive measure. In this study, nursing students’ distress disclosure was positively correlated with PTG, but did not have a direct effect on PTG. Kim and Ahn [64], reporting on the relationship between distress disclosure and PTG, noted that self-disclosure could appear differently depending on emotional regulation patterns. According to another study [65], although self-disclosure does not directly affect PTG, expressing trauma-related thoughts and emotions affects adaptation [65]. Since self-disclosure can affect PTG when accompanied by social support [66], while self-disclosure itself does not contribute to PTG, it is important in interpersonal relationships [67]. Therefore, it is necessary to clearly perceive one’s own feelings and increase the chance of self-disclosure to receive support from close others [68]. In this respect, a follow-up study is considered necessary to determine factors influencing PTG, including social support variables.

The limitation of this study is its generalizability. Caution is warranted in interpreting its results by generalizing them for all nursing students because the analysis data was collected from nursing students in one region. Therefore, it is suggested to conduct repeated studies on factors affecting PTG by expanding data collection to various regions. Nevertheless, this study is significant in that it determined the factors influencing PTG in nursing students who experienced a national disaster called the COVID-19 pandemic.

## 5. Conclusions

The significance of this study is twofold. First, it determined the factors influencing PTG in nursing students by examining the correlations between PTG (dependent variable) and emotional regulation, resilience, and distress disclosure (major independent variables). Second, it prepared basic data for establishing measures to promote PTG in nursing students.

Therefore, in order to improve nursing students’ PTG, it is necessary to help them increase their satisfaction with their major, improve their interpersonal skills, and better adapt to the clinical practice settings. Moreover, lecturers need to seek various teaching methods to motivate students to participate more actively in learning activities. It is also necessary for nursing students to set up a PTG program that can help them perceive their emotions more clearly and accurately, utilize cognitive reappraisal as a positive emotion regulation strategy, and increase resilience.

## Figures and Tables

**Table 1 ijerph-20-02782-t001:** Differences in post-traumatic growth by general characteristics (N = 231).

Characteristics	Categories	*n* (%)	Mean ± SD	t/Z/F	*p* Scheffé
Grade	Junior	115 (49.8)	3.94 ± 0.89	0.63	0.530
	Senior	116 (50.2)	3.86 ± 0.93		
Gender	Male	35 (15.2)	3.81 ± 1.05	−0.66	0.511
	Female	196 (84.8)	3.92 ± 0.88		
Transfer student	Yes	13 (5.6)	4.43 ± 0.80	−2.14	0.032 †
	No	218 (94.4)	3.87 ± 0.91		
Housing type	Home	151(65.4)	3.86 ± 0.91	−0.97	0.333
	Independent residence	80 (34.6)	3.98 ± 0.90		
Religion	No	157 (68.0)	3.85 ± 0.91	−1.13	0.258
	Yes	74 (32.0)	4.00 ± 0.91		
Economic status	High	54 (23.4)	4.02 ± 0.88	1.65	0.194
	Middle	123 (53.2)	3.93 ± 0.80		
	Low	54 (23.4)	3.90 ± 0.91		
Subjective health status	Healthy ^a^	106 (45.9)	3.99 ± 0.95	3.37	0.036 a > c
	Average ^b^	107 (46.3)	3.90 ± 0.85		
	Bad ^c^	18 (7.8)	3.40 ± 0.88		
Satisfaction with major	Satisfied ^a^	108 (46.8)	4.11 ± 0.92	8.28	<0.001 a, b > c
	Neutral ^b^	104 (45.0)	3.80 ± 0.80		
	Dissatisfied ^c^	19 (8.2)	3.30 ± 1.07		
Satisfaction with academic achievement	Satisfied	27 (11.7)	4.16 ± 0.92	2.50	0.085
	Neutral	108 (46.8)	3.96 ± 0.82		
	Dissatisfied	96 (41.6)	3.76 ± 0.99		
Satisfaction with hybrid-learning class	Satisfied ^a^	87 (37.7)	4.06 ± 0.98	4.13	0.017 a > c
	Neutral ^b^	109 (47.2)	3.90 ± 0.83		
	Dissatisfied ^c^	35 (15.2)	3.54 ± 0.89		
Satisfaction with interpersonal relationship	Satisfied ^a^	85 (36.8)	4.20 ± 0.86	12.70	<0.001 a, b > c
	Neutral ^b^	122 (52.8)	3.83 ± 0.81		
	Dissatisfied^c^	24 (10.4)	3.23 ± 1.15		
Satisfaction with clinical practice	Satisfied ^a^	63 (27.3)	4.27 ± 0.86	12.63	<0.001 a > b > c
	Neutral ^b^	130 (56.3)	3.86 ± 0.86		
	Dissatisfied ^c^	38 (16.5)	3.38 ± 0.90		

† Mann–Whitney U test; ^a, b, c^: post hoc test- Scheffé

**Table 2 ijerph-20-02782-t002:** Levels of emotional regulation, resilience, distress disclosure, and post-traumatic growth (N = 231).

Variables	Min	Max	Mean ± SD
Post-traumatic growth	1.40	6.00	3.90 ± 0.91
Emotional regulation-Suppression	1.00	5.00	2.61 ± 0.92
Emotional regulation- Reappraisal	1.00	5.00	3.32 ± 0.87
Resilience	1.96	4.90	3.70 ± 0.54
Distress disclosure	1.25	5.00	3.25 ± 0.81

**Table 3 ijerph-20-02782-t003:** Correlation among emotional regulation, resilience, distress disclosure, and post-traumatic growth.

Variables	Post-Traumatic Growth	Emotion Regulation		Resilience	Distress Disclosure
		Suppression	Reappraisal		
**Post-Traumatic Growth**	**1**				
**Emotional Regulation** **-Suppression**	0.03 (0.684)	1			
**Emotional Regulation** **-Reappraisal**	0.56 (<0.001)	0.23 (<0.001)	1		
**Resilience**	0.59 (<0.001)	0.03 (0.681)	0.62 (<0.001)	1	
**Distress Disclosure**	0.18 (0.005)	−0.62 (<0.001)	0.11 (0.094)	0.32 (<0.001)	1

**Table 4 ijerph-20-02782-t004:** Influencing Factors on Post-Traumatic Growth (N = 231).

Variables	B	SE	β	t	*p*
(constant)	0.43	0.32		1.36	0.174
Resilience	0.62	0.11	0.37	5.71	<0.001
Emotional regulation -Reappraisal	0.33	0.07	0.31	5.00	<0.001
Satisfaction with clinical practice = satisfied *****	0.29	0.10	0.14	2.85	0.005
Transfer student = yes *****	0.50	0.20	0.13	2.54	0.012
Tolerance = 0.60–0.99, VIF = 1.01–1.66, Durbin-Watson = 1.97					
Adjusted R² = 0.44, R² = 0.45, F = 45.24, *p* < 0.001					

* Dummy variables had the followed referent groups: Satisfaction with clinical practice (Dissatisfied = 0), Transfer student (No = 0).

## Data Availability

The data presented in this study are available on request from the corresponding author.

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
