# Peer review of "Effects of Emotional Regulation, Resilience, and Distress Disclosure on Post-Traumatic Growth in Nursing Students"

_ijerph, 2023, doi:10.3390/ijerph20042782_

Round 1
Reviewer 1 Report
Dear authors,
The article "Effects of emotional regulation, resilience and distress disclo-sure on post-traumatic growth in nursing students" aimed to examine the PTG factors in nursing students who had experienced the trauma of the COVID-19 pandemic with the aim of preparing basic data to help nursing students better adapt.
The article has interesting results, although they are quite local. The collection method also has its limitations. The authors can improve formal issues in the main text such as paragraph length or order of presentation of arguments in the discussion and conclusion sections. As well as improving the limitations and what is concluded from the study.
Specific issues (below):
Introduction: It would be interesting to see a brief definition of the PTSD.
Results: It’s missing the lower line in the Table 4.
Discussion: The 1st paragraph would be improved with a brief description of the main goals and findings of the study.
The subsequent paragraphs are too long for the reader. Changing the formal structure of these paragraphs would be helpful for the clear understanding of the discussion.
The limitations are not well addressed. What about the sample size? The inherent bias of the data collection method used.
Conclusion: The first half of the 2nd paragraph seems more appropriated for the 1st paragraph of the discussion section as it summarizes the main findings.
The conclusion section would present the conclusions of the present study considering the study’s goals, results, and discussion.
Reviewer 2 Report
1.Include the concept of hybrid learning in the introduction
2.Explain the recruitment of students, the inclusion criteria and wheter they were classified by the course they were studying?
3. Explain what hibrid learning consists of and how did you classify the students
Reviewer 3 Report
REVIEW
Effects of emotional regulation, resilience and distress disclosure on post-traumatic growth of nursing students
Dear Authors, I found it very interesting to read your manuscript. Your topic is significant in the current world that is not free from risks and threats of pandemics, emphasizing the role of competent nurses.
I’ll suggest some structural changes to your manuscript to meet the publication criteria of IJERPH.
Introduction
The background and some important pieces of earlier research to justify your research objectives are well presented, but a larger literature review is missing. Or, it is not missing, but rather presented in Discussion in a connection with your results. I suggest that you rewrite your paper according to the traditional standards of academic research and present all relevant earlier research in Introduction. In that way the reader will understand, what is already known.
Materials and Methods
This section is clearly written and the methods chosen are applicable in a descriptive research setting.
Results
I suggest that you will follow the traditions of scientific writing throughout this section. This means writing a short introduction, one or more sentences, to tell the reader, what results you are presenting in Table X, after which comes Table X and after Table X a description and preliminary interpretation of it.
Discussion
The traditional content of Discussion is to interpret and describe the significance of your results in relation of what was already known about your research problem. That would mean returning to the literature presented in Introduction, instead of presenting totally new references about earlier research. Limitations are presented shortly, but recommendations for further research are missing.
Conclusion
The conclusions are correctly drawn from the results and include also concrete and relevant practical implications.
I wish you all the best in your revision work and hope to see this manuscript as a published research article.
Sincerely, your reviewer
Round 2
Reviewer 2 Report
Changes have been made that have improved the article.